# Competition between lysogenic and sensitive bacteria is determined by the fitness costs of the different emerging phage-resistance strategies

Olaya Rendueles*, Jorge AM de Sousa, Eduardo PC Rocha

Institut Pasteur, Université Paris Cité, CNRS UMR3525, Paris, France

**Abstract** Many bacterial genomes carry prophages whose induction can eliminate competitors. In response, bacteria may become resistant by modifying surface receptors, by lysogenization, or by other poorly known processes. All these mechanisms affect bacterial fitness and population dynamics. To understand the evolution of phage resistance, we co-cultivated a phage-sensitive strain (BJ1) and a polylysogenic *Klebsiella pneumoniae* strain (ST14) under different phage pressures. The population yield remained stable after 30 days. Surprisingly, the initially sensitive strain remained in all populations and its frequency was highest when phage pressure was strongest. Resistance to phages in these populations emerged initially through mutations preventing capsule biosynthesis. Protection through lysogeny was rarely observed because the lysogens have increased death rates due to prophage induction. Unexpectedly, the adaptation process changed at longer time scales: the frequency of capsulated cells in BJ1 populations increased again because the production of the capsule was fine-tuned, reducing the ability of phage to absorb. Contrary to the lysogens, these capsulated-resistant clones are pan-resistant to a large panel of phages. Intriguingly, some clones exhibited transient non-genetic resistance to phages, suggesting an important role of phenotypic resistance in coevolving populations. Our results show that interactions between lysogens and sensitive strains are shaped by antagonistic co-evolution between phages and bacteria. These processes may involve key physiological traits, such as the capsule, and depend on the time frame of the evolutionary process. At short time scales, simple and costly inactivating mutations are adaptive, but in the long term, changes drawing more favorable trade-offs between resistance to phages and cell fitness become prevalent.

**\*For correspondence:**
olaya.rendueles-garcia@pasteur.
fr

**Competing interest:** The authors declare that no competing interests exist.

## Editor's evaluation

The overarching question of the manuscript is important and the findings inform the patterns and mechanisms of phage-mediated bacterial competition, with implications for microbial evolution and antimicrobial resistance. The strength of the evidence in the manuscript is compelling, with a huge amount of data and very interesting observations. The conclusions are well supported by the data. This manuscript provides a new co-evolutionary perspective on competition between lysogenic and phage-susceptible bacteria, that will inform new studies and sharpen our understanding of phage-mediated bacterial co-evolution.

## Introduction

Parasites shape the life history and fitness of their hosts. They also impact community structure via predation and competition, and thereby affect numerous ecological and evolutionary processes

(*Koskella and Brockhurst, 2014*; *Pedersen and Fenton, 2007*; *Lefèvre et al., 2009*). Bacteriophages (phages) are very abundant predators of bacteria (*Brüssow and Hendrix, 2002*; *Suttle, 2007*). Temperate phages either follow a lytic cycle in which they replicate within bacterial cells and release infectious virions, or a lysogenic cycle in which they integrate the bacterial genome and replicate with it. Nearly half of the sequenced bacterial genomes are lysogens (*Touchon et al., 2016*). The dual lifestyle of temperate phages is costly, but can also provide the host with multiple advantages. During lysogeny, prophages may increase biofilm formation (*Gödeke et al., 2011*), phosphate acquisition (*Sullivan et al., 2005*), or express virulence factors (*Busby et al., 2013*; *Fasano et al., 1991*; *Varani et al., 2013*). Inactivated prophages leave genes in the genome that are co-opted by the host and result in functional innovation, e.g., as bacteriocins used in bacterial warfare (*Winstanley et al., 2009*; *Bobay et al., 2014*; *Nakayama et al., 2000*). Prophages also protect bacteria from closely related phages, a process called superinfection resistance (*Bondy-Denomy et al., 2016*). Furthermore, when the lytic cycle is initiated in a small subpopulation, it may facilitate colonization by directly mediating competition within communities (*Li et al., 2017*; *Joo et al., 2006*; *Harrison and Brockhurst, 2017*; *Wendling et al., 2021*), because the released virions will infect and lyse closely related but not identical strains. This can promote the acquisition of adaptive traits from bacterial competitors (*Wendling et al., 2021*; *Haaber et al., 2016*). Hence, it is suggested that prophage induction affects bacterial population dynamics, community structure, and evolution (*Bondy-Denomy and Davidson, 2014*; *Bossi et al., 2003*; *De Paepe et al., 2016*; *Gama et al., 2013*; *Nanda et al., 2015*).

How parasite pressure may alter co-evolving bacterial populations has been seldom addressed, and most of these studies focused on virulent phages (*Brockhurst et al., 2007*; *Fazzino et al., 2020*; *Lourenço et al., 2020*; *Weinbauer, 2004*; *Weinbauer and Rassoulzadegan, 2004*). A few other studies have tested the impact of coevolution between lysogens and non-lysogens and the advantages the former provides in vivo by mediating bacterial interactions (*Joo et al., 2006*; *De Paepe et al., 2016*; *Burns et al., 2015*; *Davies et al., 2016*; *Frazão et al., 2022*). However, the relevance of poly-lysogeny for population dynamics during hundreds of generations remains unknown. Further, little is known about the interactions between the different resistance mechanisms, how they affect the cost of resistance, and how they may provide an opportunity for the emergence of novel mechanisms. To test this, we co-evolved two natural isolates of *K. pneumoniae,* an ubiquitous species, of which at least 75% of the species' genomes are polylysogenic (*de Sousa et al., 2020*): (i) the hypervirulent BJ1 strain without inducible or cryptic prophages, that was isolated from a liver abscess (ST380) and (ii) a polylysogenic multidrug-resistant *K. pneumoniae* strain (ST14) isolated from a urinary tract nosocomial infection. ST14 produces multiple infectious virions for which the BJ1 is known to be sensitive (*Supplementary file 1*; *de Sousa et al., 2020*). Based on previous studies, cell defense mechanisms, such as restriction-modification systems, are not expected to impact population dynamics (*de Sousa et al., 2020*). We hypothesized that resistance would rather emerge by lysogenization under strong phage pressure and by inactivation of the extracellular capsule, the main surface receptor of phage at intermediate and low phage pressure (*de Sousa et al., 2020*). To test if, and how, prophage induction affects the competition outcome between the two strains, we followed their population dynamics through time. We then tested for the emergence of phage resistance in the susceptible strain. This revealed the diversity and interactions of the emerging mechanisms of phage resistance. It also provided unique insight into how these different mechanisms coexist within a population and evolve through time in response to infection pressure.

## Results

### Temperate phages provide fitness advantage during competition

We first aimed at understanding if the prophages of strain ST14 provide a fitness advantage during competition with phage-susceptible strain BJ1. To limit confounding factors such as competition for resources, we grew the cells in a rich environment. To modulate the amount of phage produced, and the ability of the latter to infect, we defined three conditions: (i) LB, (ii) LB supplemented with 0.2% citrate to inhibit phage infection due to calcium chelation (*de Sousa et al., 2020*; *Shafia and Thompson, 1964*), and (iii) LB with mytomicin C (MMC, 0.1 µg/mL) to increase the phage titers in the environment. MMC was added at a concentration that did not significantly affect the growth of BJ1 (*Figure 1—figure supplement 1A, B*), and despite the consumption of citrate by *Klebsiella*, after

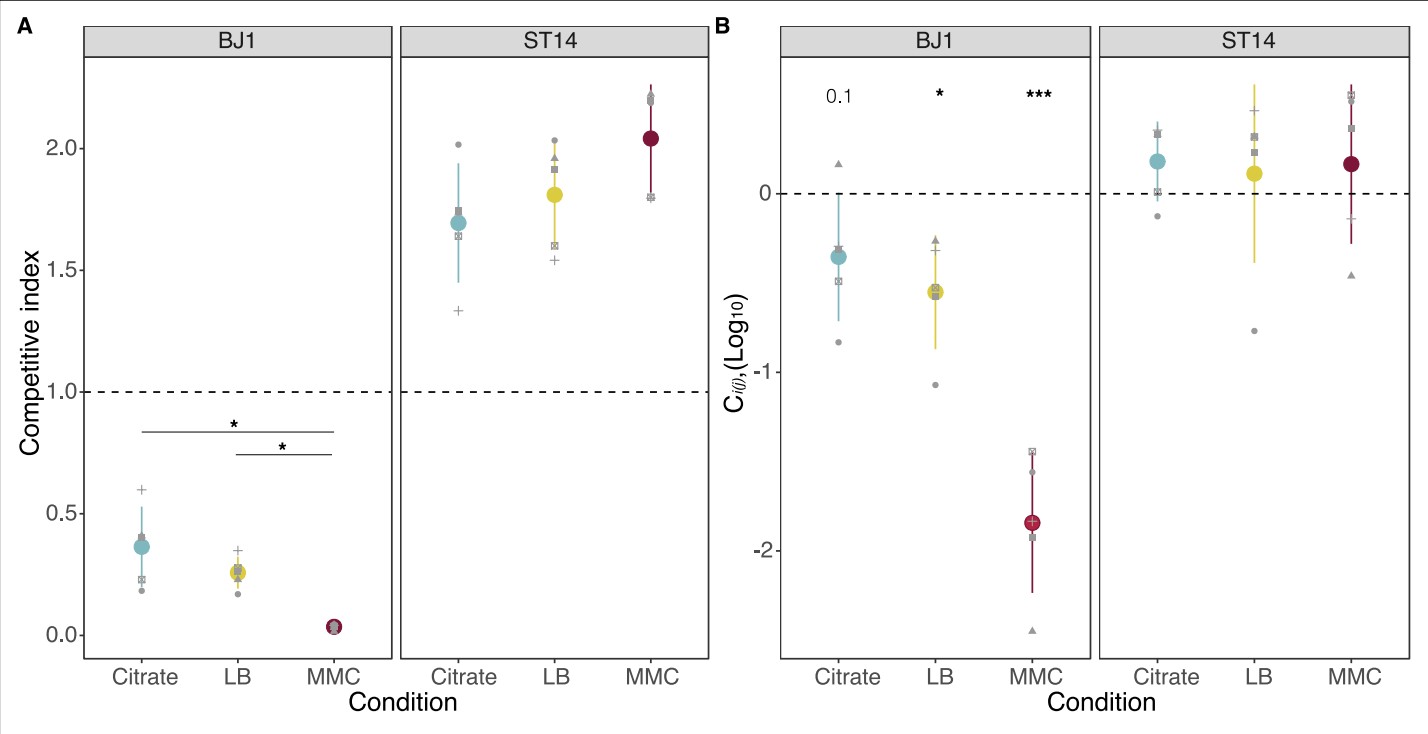

**Figure 1.** Fitness of strains during competition. (**A**) The competitive index is calculated as the final frequency of each strain divided by the initial frequency in the mixed cocultures. * $P < 0.05$, Wilcoxon rank sum test adjusted by Benjamini-Hochberg correction. (**B**) The effect of mixing two strains during growth in coculture is given as C$i(j)$, expressed in $\log_{10}$, with $i$ representing either strain BJ1 or strain ST14. Positive values represent increased cell numbers during coculture than those expected from the pure cultures. p-values correspond to a one-sample t-test for a difference of 0. *p<0.05, ***p<0.001. Each dot shape represents an independent biological replicate, N=5. Error bars indicate the standard deviation.

The online version of this article includes the following figure supplement(s) for figure 1:

**Figure supplement 1.** Growth and phage production in experimental conditions.

**Figure supplement 2.** Total growth in the different media.

24 hr there is still a large amount of citrate remaining, that is sufficient to inhibit infection (*Figure 1—figure supplement 1*). We also quantified the amount of PFU/mL produced by strain ST14 in the different growth conditions. As expected, phage production in ST14 was significantly higher in MMC compared to the two other treatments. Interestingly, ST14 grown in citrate resulted in a marginal increase in phage production relative to the control (LB) (*Figure 1—figure supplement 1C, D*).

To test whether phages could contribute to the competitive fitness of their host, we co-inoculated both strains (BJ1 and ST14) at an initial ratio of 1:1 for 24 hr. First, we tested if mixing the two strains affected the total growth or the population yield. We observed no increased cell death due to the competition in the different growth conditions (*Figure 1—figure supplement 2A*). Then, we calculated the competitive index of the strains and observed that there was a large fitness advantage for strain ST14 in all three conditions (*Figure 1A*). This was most likely because ST14 has both a higher growth rate and population yield than BJ1, except in the presence of MMC (*Figure 1—figure supplement 1A, B*). Most importantly, we also observed differences in the competitive index depending on the amount of phage released, or its ability to infect the BJ1 (Kruskal-Wallis, dF = 2, p=0.006). This effect could be due to the different population yields of each strain in each environment (*Figure 1—figure supplement 2B*). To take into account only the fitness effects due to phages, we quantified the strain-interaction effects using C$_i(j)$, which measures the effect of mixing two strains $i$ and $j$ on the viable population size of strain $i$, relative to pure culture controls. This measure accounts for the absolute performance of each competitor in mixed groups (see Methods). Negative C$_i(j)$ values indicate that strain $i$ have a lower population yield during growth in the presence of $j$ than in pure culture, and positive values indicate the opposite. For the phage producer, strain ST14, the competition had no positive or negative effect on total population yield, most likely because the increased release

of viruses resulting in ST14 death is outweighed by an exacerbated death of phage-sensitive BJ1 (*Figure 1B*). In the presence of citrate, a condition in which phage cannot infect, the growth of strain BJ1 was not significantly inhibited. However, in the absence of citrate, when phages can infect, $C_i(j)$ was significantly lower than zero, indicating a negative effect on the growth of strain BJ1. This effect was dependent on the amount of phage released into the environment, as an increased production of phages by ST14 due to MMC leads to an even lower $C_i(j)$ for BJ1 (*Figure 1B*, Kruskal-Wallis, dF = 2, p=0.007).

We also tested whether ST14 could sense the presence of competition, for instance by quorum sensing mechanisms, and induce prophages and the production of viral particles (*León-Félix and Villi-caña, 2021*). Our results showed that the growth of ST14 with spent supernatant of BJ1 did not result in increased viral release (*Figure 1—figure supplement 1C, D, E*). Taken together, our results showed that prophages can increase the fitness of their host in co-culture by disfavouring the non-lysogens.

## Resistance to temperate phages emerges rapidly during coevolution

To assess whether ST14 prophages could provide a long-term fitness advantage to their hosts, and allow them to outcompete non-lysogens, we set up an experiment in which we allowed three independent mixed populations composed of phage-producing ST14 and phage-susceptible BJ1 strains to coevolve during 30 days, in the three previously defined environments (LB, LB supplemented with 0.2% citrate, and LB supplemented with MMC). To follow the evolution of each strain, we plated the populations every day on selective media and counted CFU. As expected, no significant changes in the group yield were observed (*Figure 2—figure supplement 1*). This is mostly explained because the dominant strain, the phage producer, does not change its population yield (*Figure 2A*). In contrast, the frequency of BJ1 decreased rapidly during the first four days, suggesting a large initial fitness disadvantage of this strain. This is observed in all three conditions, but it is accelerated in conditions in which phage release is exacerbated (with MMC) and bacterial infection is not restricted (without citrate). Decrease in BJ1 populations was also correlated with large increases in phage production during the first ten days (*Figure 2—figure supplement 2*). Interestingly, shortly after the beginning of the experiment, both phage production and evolved BJ1 populations seem to stabilize, except for one BJ1 population evolving in MMC (which increases significantly in frequency). This suggested the emergence of phage resistance. However, after day 15, BJ1 populations evolving in MMC remain stable whereas the ones in the other conditions showed a second decrease in CFUs. This second decrease continued until day 22, beyond which BJ1 populations once again stabilized at *ca* $10^3$ CFU/mL. Taken together, in 30 days of coevolution, ST14 did not completely displace BJ1 from the populations, even in conditions where the production of the phage by strain ST14 is exacerbated. Indeed, across all conditions, and throughout all the experimental evolution, infectious virions were actively produced and released (*Figure 2—figure supplement 2*), representing a constant, active selective pressure. This suggests that prophage-mediated competition can be counter-balanced by the evolution of resistance mechanisms in the competitor strain.

## Phage pressure drives receptor inactivation as a mechanism of resistance

It has been largely documented that the extracellular capsule is a main phage receptor in *Klebsiella* (*de Sousa et al., 2020*; *Hesse et al., 2020*; *Tan et al., 2020*; *Venturini et al., 2020*). Throughout the daily plating of coevolving populations, we observed the rapid emergence of non-capsulated clones in all independent populations across the three treatments (*Figure 2B*). We tested if non-capsulated clones could be under stronger selection for receptor inactivation when phage pressure is higher (higher density of phages). We observed that the emergence of non-capsulated clones in the BJ1 background is exacerbated in the environment in which phage pressure is greater (insert, *Figure 2B*), and is diminished when phages cannot infect. Hence, phage pressure accelerated capsule inactivation. Interestingly, this is also the case for ST14, the phage producer, which we had previously shown to be mildly susceptible to its own phages (*de Sousa et al., 2020*). Overall, within the first ten days, at least 50% of the population was composed of non-capsulated mutants. Towards the end of the experiment, increasing frequencies of capsulated clones were observed across many populations, suggestive of the emergence of other resistance mechanisms.

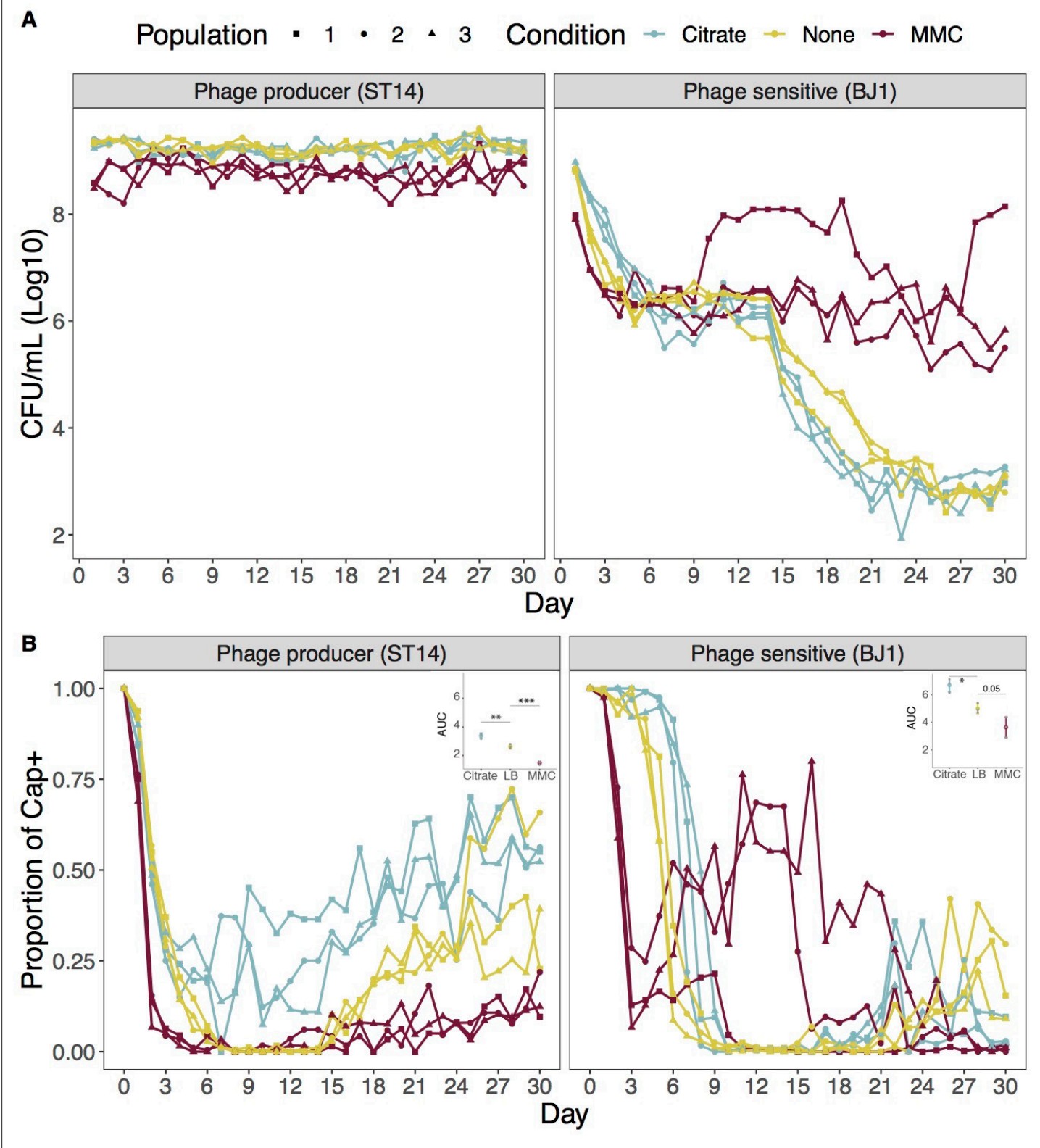

**Figure 2.** Population yield and proportion of capsulated clones of the two strains during the coevolution experiment. (**A**) Total CFU per mL of each strain was estimated every day on selective media. Each line represents an independent coevolving population. (**B**) Emergence of non-capsulated mutants in each strain. The proportion of capsulated clones in the population is depicted. The insert shows the area under the curve (AUC) during the

*Figure 2 continued on next page*

*Figure 2 continued*

first nine days of evolution, as calculated by the function *trapz* from the R package pracma. *p<0.05,**p<0.01,***p<0.001 for ANOVA with Tukey *post hoc* corrections.

The online version of this article includes the following figure supplement(s) for figure 2:

**Figure supplement 1.** Total number of cells across evolving populations.

**Figure supplement 2.** Estimated phage titers in each population throughout the coevolution experiment.

We sequenced the gene *wcaJ* to identify the genetic causes of receptor inactivation because it encodes the first glycosyltransferase of the capsule biosynthesis pathway and is known to be largely responsible for capsule inactivation (*Chiarelli et al., 2020*; *Haudiquet et al., 2021*). This revealed that all but two non-capsulated clones (BJ1 (N=36); ST14 (N=18)) had mutations in *wcaJ*, most of which resulted in a loss-of-function (*Supplementary file 2*). In summary, phage pressure led to rapid resistance emergence by surface modifications.

### The emergence of new lysogens is rare and potentially unstable

Some resistant clones of strain BJ1 are capsulated, which led us to hypothesize that they evolved other resistance mechanisms. To test this, we analyzed at different time points the resistance mechanisms of the capsulated clones in the population. We expected to find BJ1 lysogens, since super-infection exclusion due to the lysogenization of capsulated bacteria could prevent further infection by the same phages. Further, our previous work had already shown that, when infected with phage lysate at high titers, at least two of the four intact phages from strain ST14 could lysogenize BJ1 (*Supplementary file 1*; *de Sousa et al., 2020*). To quantify the proportion of lysogenized BJ1 cells, relative to other resistance mechanisms, we isolated over 1200 capsulated clones at different time points (*Figure 3—figure supplement 1*). We identified the clones that were resistant to purified phage lysates of strain ST14 and that produced phages when exposed to MMC in our culture conditions. More precisely, we analyzed the differences in the area under the growth curve of each clone, both when they were grown in LB (control), when phage lysate was added (to distinguish between resistant or susceptible), and when MMC was added (to induce prophages and identify newly lysogenic clones). Together with the resistant non-capsulated clones (*Figure 2B*), this provides a detailed overview of the different mechanisms of resistance, their proportion, and their temporal dynamics throughout the experiment (*Figure 3*).

We first observed that the proportion of susceptible clones quickly decreased, especially when phage pressure was high (under MMC, *Figure 3*). The majority of tested clones were resistant by day 2, 6, and 8 in populations evolving in MMC, LB, and citrate, respectively. As expected, lysogens emerged in all populations, but remained at low frequency and their numbers quickly dwindled after their emergence (*Figure 3*). We verified that the 94 identified lysogens, out of the 1209 screened clones, were *bona fide* lysogens. This could be confirmed by their production of phages infecting naïve BJ1 cells, both when induced by MMC (92 out of 94), and in the absence of induction (87 clones out of 94) (*Figure 3—figure supplement 2*).

Interestingly, we observed that when new lysogens are grown in LB, in the absence of induction, there is a detectable amount of cell death, and growth delay at the end of the exponential phase in at least in 29 out of 94 tested clones (*Figure 3B* and *Figure 3—figure supplement 4*). This could correspond to a high frequency of spontaneous induction in the newly lysogenized bacteria. Indeed, we observed a large amount of phage release, as evidenced by large inhibition halos on an overlay of ancestral BJ1. We then selected five different lysogens that consistently showed large inhibition halos (*Figure 3—figure supplement 2*) and exacerbated death (*Figure 3B* and *Figure 3—figure supplement 3*) descending from three independently evolving BJ1 populations. We quantified the amount of phage released, in the absence of induction, on a lawn of ancestral BJ1. New lysogens produced between 100 and 1000 more PFU/mL than the ancestral phage producer (ST14) (*Figure 3C*). This suggests that protection by lysogeny results in significant fitness costs because prophage induction is frequent (*Figure 3—figure supplements 2 and 3A*).

To study the impact of prophage acquisition on the long-term stability of lysogens in a population, we used eVIVALDI, an individual-based model for microbial interactions and evolution (*de Sousa and Rocha, 2019*). We used these simulations to explore different rates of induction of prophages, in the

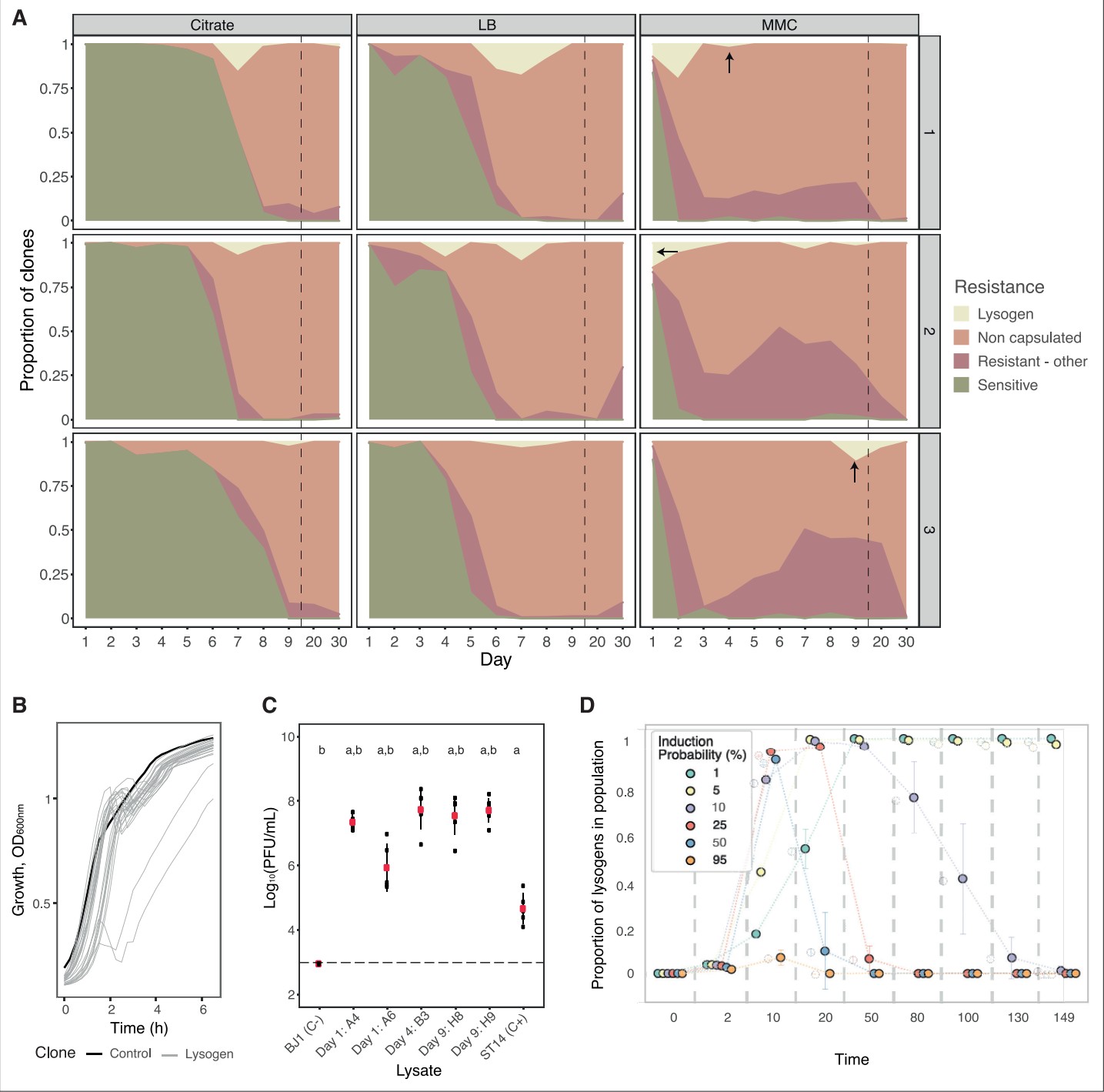

**Figure 3.** Evolution of resistance mechanisms in strain BJ1. (**A**) Ratio of clones from each coevolving population that are susceptible (green), non-capsulated (light pink), capsulated lysogens (beige), or capsulated but resistant by other undefined mechanisms (dark pink). N.B. Dashed line indicates when the x-axis, no longer follows a linear scale. Dark arrows indicate the time points from which the lysogens tested in panel C were retrieved. (**B**) Growth of newly lysogenized clones reveals significant death during the exponential phase (in the absence of induction), as measured by the optical density. Black line corresponds to the control, BJ1 ancestor. (**C**) PFU/mL produced without induction by five selected new lysogens derived from BJ1 and isolated at day one for A4 and A6 (Population #2, MMC), at day four for B3 (Population #3, MMC), and day nine for H8 and H9 (Population #3, MMC). Dashed line indicates the limit of detection of the essay. Each black dot represents an independent biological replicate (independent strain lysate) and large red dots represent the mean. Error bars correspond to the standard deviation. Two-sided t-test 'a', p<0.001 compared to ancestor BJ1 (negative control, C-) and 'b', p<0.05 compared to ST14 (phage producer, positive control, C+). (**D**) Simulated temporal dynamics of the proportion of lysogens in the populations, as calculated by eVIVALDI. Each circle corresponds to the central tendency of replicate simulations, with the different colors

*Figure 3 continued on next page*

*Figure 3 continued*

indicating a given probability of spontaneous prophage induction (shown in the legend, values approximated to the nearest major integer). The error bars correspond to the standard deviation across the replicate simulations. In the represented simulations, the probability of acquisition of a phage resistance mutation (capsule loss) is 0.001, and the fitness cost of this mutation is 10% of the bacterial growth rate, as calculated in *Buffet et al., 2021*.

The online version of this article includes the following figure supplement(s) for figure 3:

**Figure supplement 1.** Number of clones from strain BJ1 analyzed each day from each population.

**Figure supplement 2.** Lysogens can release viable phages that infect the ancestral BJ1 strain.

**Figure supplement 3.** Growth of newly lysogenized clones.

**Figure supplement 4.** Simulated temporal dynamics of the competition between different phage resistance mechanisms.

presence or absence of abiotic agents. We designed a scenario where a population of initially sensitive bacterial cells is exposed to an inoculum of temperate phages, and we followed the populations for a period of 150 iterations (e.g. approximately 150 generations). Simulated bacteria could either be infected by phage (thus either dying upon a lytic infection or becoming lysogens if the phage integrates the bacterial genome) or become resistant to phage by mutation (i.e. capsule inactivation, which decreases their growth rate). We then measured, over time, both the total number of cells and the proportions of lysogens.

We observed two main patterns. When prophages have low spontaneous induction rates (1 to 5%), they generate stable, non-costly lysogens. Consequently, phages spread slowly in the bacterial population, which gives time for the phage-resistant mutants to emerge and increase to high frequencies. However, because these mutations are costly, lysogens slowly displace these mutants. This results in a sigmoidal-like temporal frequency of lysogens, where at the end of the simulations most of the resistant population is composed of lysogens (*Figure 3D* and lower left part of panel *Figure 3—figure supplement 4*). These dynamics are in contrast with the bell-shaped dynamics observed for high or intermediate rates of spontaneous prophage induction (i.e. ≥11%), where lysogens quickly invade the population but are absent at the end. Such high rates correspond to unstable lysogens that quickly die due to spontaneous induction of their prophages. These conditions facilitate the propagation of phage throughout the population (due to fast phage amplification), and result in the rapid emergence of new bacterial lysogens (t=10 in *Figure 3D* and *Figure 3—figure supplement 4A*). Becoming a lysogen can be advantageous if these cells are protected from new phage infections. However, if there are high rates of prophage induction, lysogeny may become less adaptive than other mechanisms of protection, e.g., receptor loss (*Figure 3—figure supplement 4B*). As a result, when spontaneous induction rates are high and other, potentially fitter, resistant clones emerge, few or no lysogens are expected to survive as they will be outcompeted (top-right areas for the heatmaps in *Figure 3—figure supplement 4A*). This is consistent with our experimental results, where BJ1 clones quickly become lysogens with high induction rates which leads to their removal from the population by the end of the experimental evolution.

In our simulations, the absence of capsule-inactivating mutations (resistance probability = 0, right-most column of the heatmaps in *Figure 3—figure supplement 4A*), implies that populations either become extinct (if induction rates are too high) or are completely composed of lysogens. In contrast, our in vitro experiments revealed some resistant clones that were still capsulated and non-lysogens, indicating alternative mechanisms of resistance to phage. These novel clones were more frequent in populations under high phage induction pressure (MMC), when the cost of lysogeny is high, and less frequent undergrowth in LB with or without citrate (Kruskal-Wallis, dF = 2, *P*=0.03) (*Figure 3*). Taken together, our results show that most clones became resistant to capsule inactivation, a few by lysogenization, and others by novel mechanisms. These experimental results fit our simulations in suggesting the existence of competition between multiple phage resistance mechanisms.

## The cost of the resistance mechanism varies with phage pressure and across time scales

To test the competitive fitness of the evolved clones having different phage-resistance mechanisms, we compared the area under growth curves of all BJ1 evolved clones isolated in the presence or in the absence of phage (*Figure 4*). In the first ten days of the evolution experiment, non-capsulated clones had a higher AUC than all other resistant clones in the presence of the phage and in the controls. This

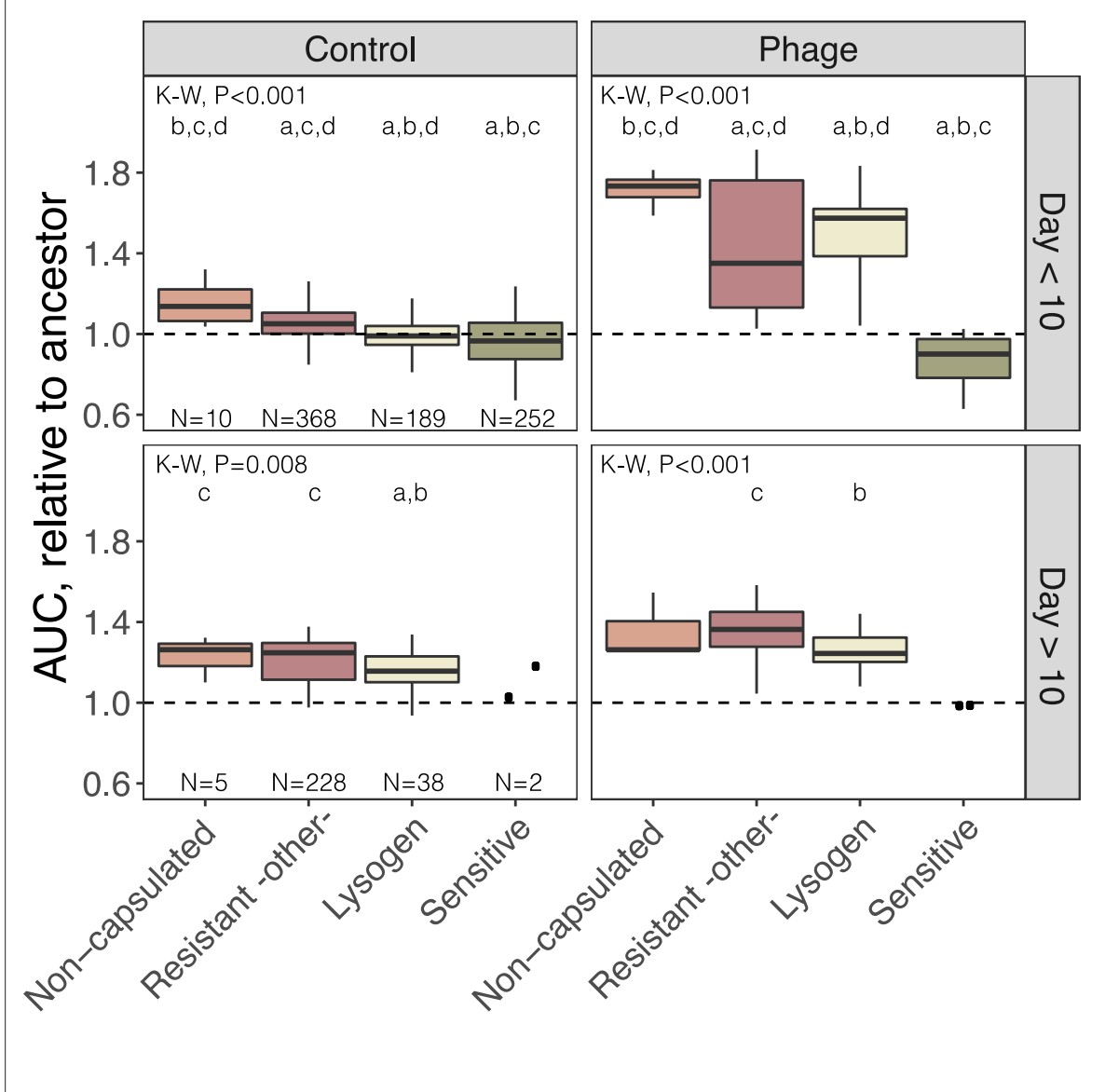

**Figure 4.** Growth of BJ1-resistant clones. All isolated clones that were capsulated and tested for resistance (*Figure 3A* and *Figure 3—figure supplement 1*) were grown in the absence of phage (control) or with phage. One or two non-capsulated clones per day were randomly selected and included in the analyses. The area under the growth curve of each clone was estimated, and compared to that of the BJ1 ancestor (dashed line). K-W, Kruskal-Wallis test. *Post hoc* tests for significant differences across groups were calculated and p-values adjusted for multiple testing with Bonferroni correction. *a*, p<0.05 difference from non-capsulated; *b*, p<0.05 difference from resistant by other mechanisms; *c*, p<0.05 difference from lysogens; *d*, p<0.05 difference from sensitive strains.

can be explained both by their intrinsic fitness advantage in nutrient-rich environments, compared to wild-type clones (*Buffet et al., 2021*) and by their efficient resistance to phage (*de Sousa et al., 2020*; *Hesse et al., 2020*; *Tan et al., 2020*). Hence, their fitness advantage likely drove their rapid expansion in the population (*Figure 3*).

AUC analyses also revealed that lysogens are less fit than the resistant capsulated clones, in the absence of phage. However, this is not so, in the presence of phage at short evolutionary time scales. At longer time scales (after ten days of evolution), resistant capsulated clones are fitter than lysogens, in the presence of phage, supporting our previous observations that lysogeny incurs a high fitness cost. This suggests that resistant mechanisms that emerge more frequently or that are accessible in evolutionary terms, such as capsule inactivation and lysogeny could be initially selected for, but become less advantageous at longer timescales where other less costly resistance mechanisms seem

to provide higher fitness. Overall, our results show a hierarchy in the competitive advantages of phage resistance mechanisms, that varies across time scales and phage pressure.

## Several changes in the receptor production play a role in resistance to phages

To identify the mechanisms of resistance to phages that involved neither receptor inactivation nor lysogeny, we characterized twelve random clones out of the 328 clones with such profiles. We measured their capsule production and resistance to purified phage lysate, either on a layer of melted agar or during growth in liquid culture. We then tested the ability of each clone to adsorb phage lysate to understand if resistance occurs prior to entering the cell. As controls, we used the ancestral strain (BJ1), as well as an Δ*rcsB* mutant, with reduced capsule expression, and a non-capsulated Δ*wcaJ* mutant (*Figure 5*). Additionally, we performed whole genome sequencing on all twelve resistant clones and looked for mutational targets, using the ancestral sequence as a reference (*Table 1*).

The integration of these analyses revealed several resistance genotypes. Five independent clones had mutations in the capsule operon. Two clones (3E1 and 9G11) had frameshift mutations in a gene coding for an acyltransferase, *orf13*, and potentially leading to a change in the capsule's biochemical composition (*Figure 5A*, *Table 1*). These clones were fully resistant to phage both in liquid and on agar and had a diminished capsule production, comparable to the phage-susceptible Δ*rcsB* mutant. However, phage particles could not successfully adsorb to the surface (*Figure 5B*). The three remaining clones had a non-synonymous mutation in *orf13* (2D2) and in *wcaJ* (9H3) and an 11 base-pair deletion in the capsule regulator *wzi* (9H7). These clones have reduced capsule expression comparable to mutations 3E1 and 9G11 in *orf13*, reduced phage adsorption, and an increased resistance to phage in liquid media (*Figure 5B*). Surprisingly, these three clones are susceptible to phage when growing on agar. These results suggest that the effect of small capsule modifications on phage resistance might be dependent on the environment. Finally, we found no mutations in known phage defense mechanisms, such as CRISPR-Cas or restriction-modification enzymes. Taken together, our results show that there are multiple paths to resistance that involve modulating either the capsule amount or its composition.

Interestingly, the remaining seven clones that were identified in our initial screens to be resistant seem to be susceptible to phage lysate in all subsequent tests. Despite their marginally lower capsule production, we could not detect mutations in their genome relative to their ancestors (except for one clone with an intergenic mutation). To discard the possibility that this could be due to a problem in our initial screen for resistant clones, we returned to the original glycerol stocks and retested these clones for their resistance to phage during growth in liquid (i.e. the same conditions as the screen) (*Figure 6*). To avoid a possible loss of the phenotype due to culture passaging, we initiated the growth curves directly from the glycerol stock without performing a preconditioning culture, that is, an acclimation step. When the culture reached OD ~0.2, we added the phage lysate. We observed that the cultures grown directly from the stocks were resistant to phage. The difference between the clones from the glycerol stock and those sequenced is that the sequenced clones underwent two extra rounds of LB passaging without phage pressure. Hence, these results suggest that transient resistance to phages can emerge without mutations (*Figure 6* and *Figure 6—figure supplement 1*).

## Receptor modifications but not lysogenization provide cross-resistance to other phages

We sought to test whether resistance to phages from ST14 could result in resistance to phages produced by other strains. To test potential cross-resistance between lysates, we produced phage lysates from strains 03–9138, ST17, and T69, all of which were previously shown to successfully infect BJ1 (*de Sousa et al., 2020*). We first infected the non-lysogenized BJ1 clones that are capsulated and resistant to ST14 phages. We observed that these clones were also resistant to the phage in lysates of strain 03–9138 and ST17 but not those of T69, suggesting that one of the phages of T69 may not use the capsule as a primary receptor (*Figure 7A* and *Figure 7—figure supplement 1*). Cross-resistance could result from phages sharing a specificity for the capsule serotype (*Beamud et al., 2023*). The strains from which we produced the lysate have very dissimilar prophages, as determined by wGRR, a measure of phage similarity for all intact phages –see Methods- (wGRR <0.25, *Figure 7—figure supplement 1B*, *de Sousa et al., 2020*). Yet, 23 proteins from these phages showed sequence identity higher than 50% with proteins present in the two ST14 phages (*Figure 7—figure supplement*

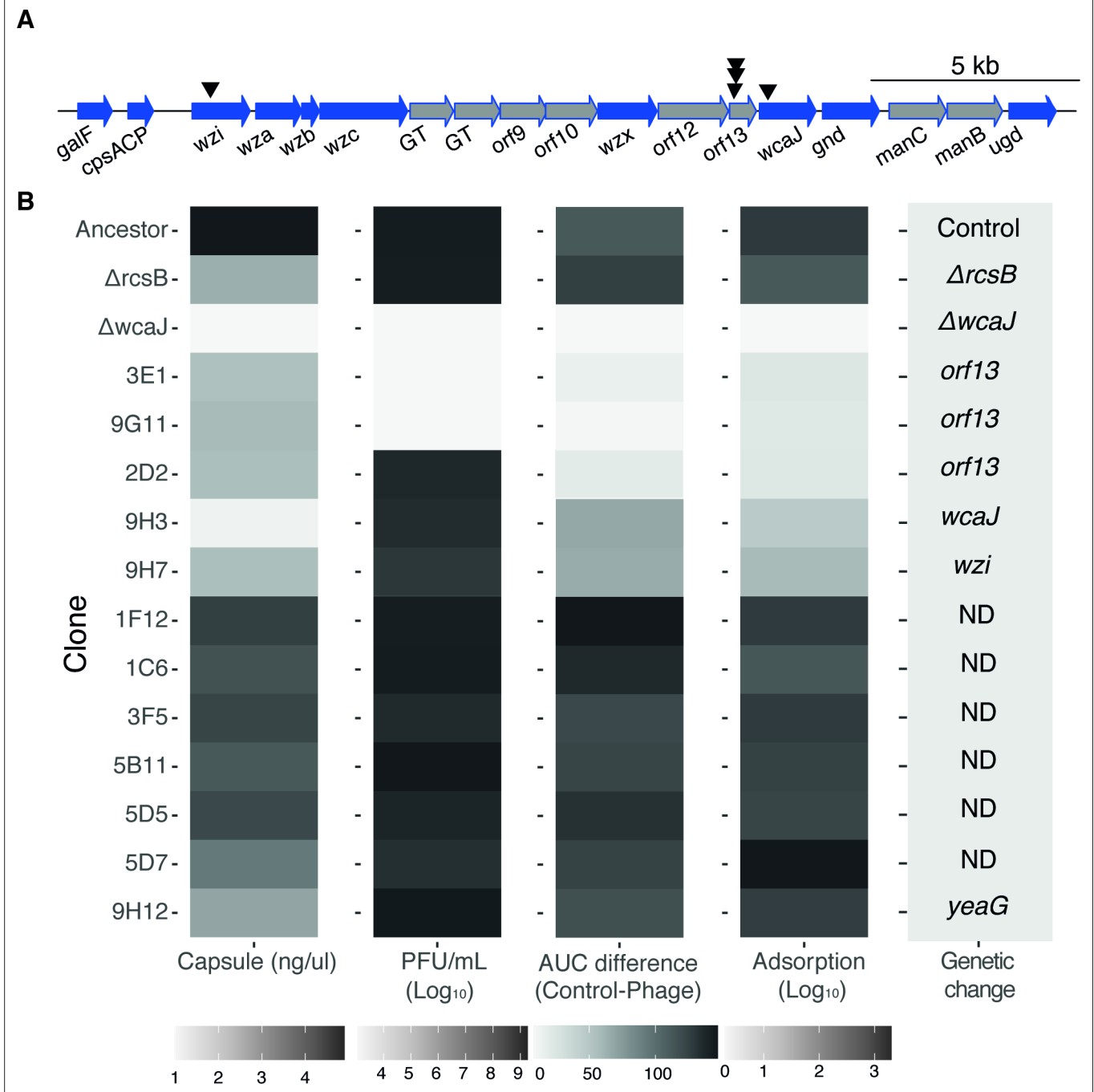

**Figure 5.** Characteristics of phage-resistant clones. (**A**) Schematic organization of the capsule operon of strain BJ1. Black triangles represent mutations observed in the capsule operon. Blue arrows indicate core genes common to all *K. pneumoniae* capsule serotypes. Gray arrows correspond to serotype-specific genes. GT stands for glycosyltransferase. The diagram was generated with genoplotR package. (**B**) For each clone, we evaluated the amount of capsule produced, the ability of the phage to be adsorbed, and phage the sensitivity on overlay by EOP (PFU/mL) and on liquid culture by AUC. The area under the curve (AUC) difference accounts for the surplus of growth of the control compared to growth with phage. If the clone is resistant, the difference in AUC is ~0. The average of three independent replicates is shown. The experiments were performed with three independently generated lysates, when applicable. ND: none detected.

1B). The functional analyses of these proteins using pVOG revealed that some are structural proteins potentially involved in infection (e.g. tail proteins) (*Figure 7—figure supplement 1D*). They may provide different phages with similar tropism, thereby explaining the observed cross-resistance among lysogens of the same serotype.

**Table 1.** List of mutations identified in the resistant clones sequenced.

Location indicates if the mutation is found on the chromosome (C) or plasmid (P). Pop stands for the population. The number of mapped sequences is also reported. Clones with less than 98% of mapped sequences are displayed in italics.

| Clone | Pop | Condition | Day | Location | Position | Change | Mutation type | Annotation | Function | Mapped sequences |
|---|---|---|---|---|---|---|---|---|---|---|
| 1 C6 | 1 | MMC | 1 | | | | | | | 98.3 |
| 1 F12 | 2 | MMC | 1 | | | | | | | 98.4 |
| 2D2 | 2 | MMC | 2 | C | 3,994,990 | T→A | I110F (ATT→TTT) | *orf13* | K2 capsule gene; Polysialic acid O-acetyltransferase | 98.2 |
| 3E1 | 1 | MMC | 3 | C | 3,995,203 | $(T)_{8\to7}$ | coding (115/669 nt) | *orf13* | K2 capsule gene; Polysialic acid O-acetyltransferase | 98.4 |
| 3 F5 | 3 | MMC | 3 | P | 34,553 | A→G | intergenic (+401/−389) | | hypothetical protein/hypothetical protein | 98.6 |
| 5B11 | 1 | LB | 5 | P | 34,561 | A→G | intergenic (+409/−381) | | hypothetical protein/hypothetical protein | 98.2 |
| 5D5 | 2 | LB | 5 | | | | | | | 98.5 |
| 5D7 | 2 | LB | 5 | | | | | | | 98.6 |
| 9 G11 | 1 | LB | 9 | C | 3,995,203 | $(T)_{8\to9}$ | coding (115/669 nt) | *orf13* | K2 capsule gene; Polysialic acid O-acetyltransferase | 98.3 |
| | | | | C | 2,890,708 | C→T | R121R (CGG→CGA) | *dcuR* | Transcriptional regulatory protein DcuR | |
| 9 H7 | 2 | Citrate | 9 | C | 4,009,965 | Δ11 bp | coding (245-255/630 nt) | *wzi* | K2 regulatory capsule gene | 98.5 |
| 9 H3 | 1 | Citrate | 9 | C | 3,994,389 | C→T | M61I (ATG→ATA) | *wcaJ* | UDP-glucose: undecaprenyl-phosphate glucose-1-phosphate transferase | 98.5 |
| | | | | C | 2,575,329 | C→T | D615N (GAC→AAC) | *yeaG* | Protein kinase YeaG | |
| 9 H12 | 3 | Citrate | 9 | P | 23,432 | G→A | R34Q (CGG→CAG) | *soj* | Phage cox protein (PF10743), annotated as plasmid-partitioning | 98.6 |

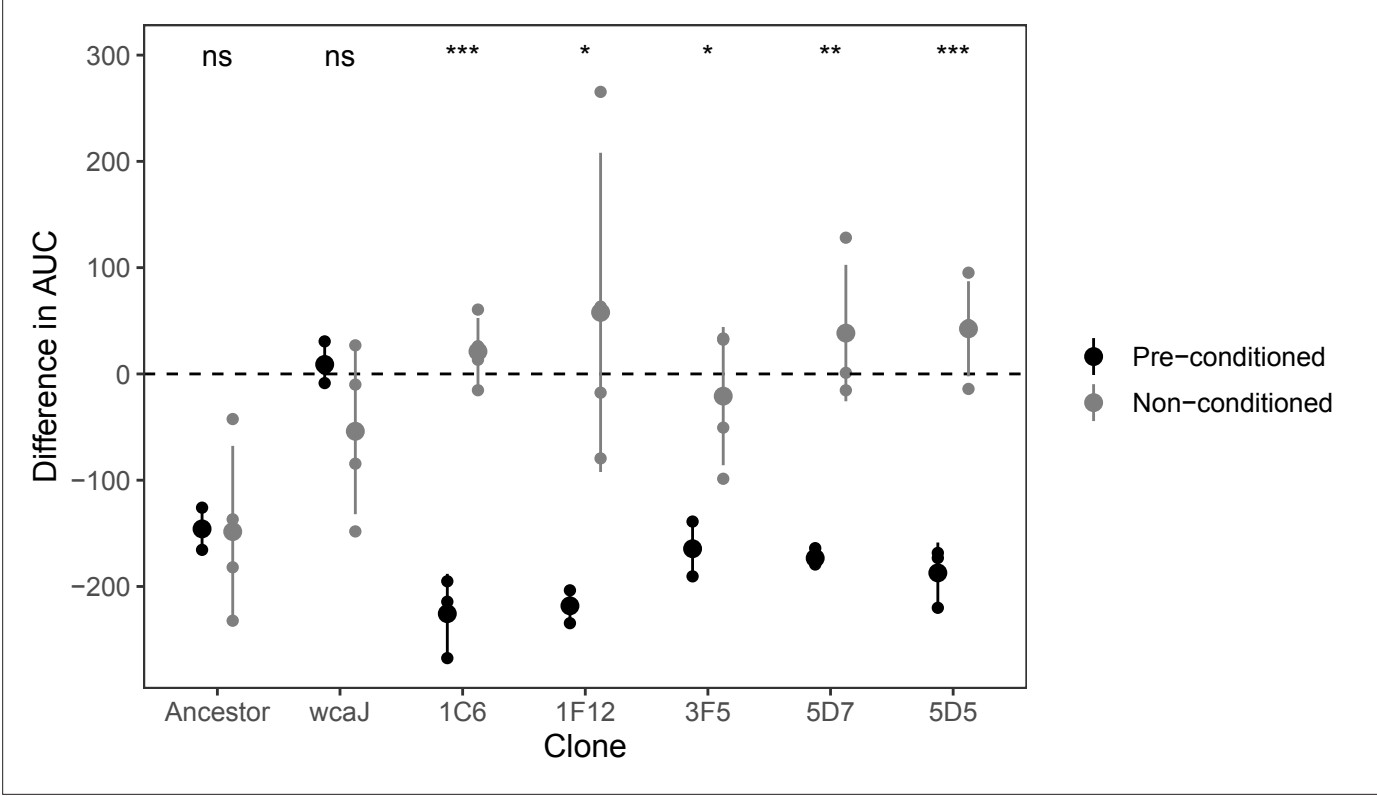

**Figure 6.** Transient resistance to phages. The difference in the area under the curve (AUC) represents the effect of adding phage to a growing culture as estimated by the difference in the growth curve in the absence of phage and with phage (*Figure 6—figure supplement 1*). Values below 0 indicate strains are sensitive to phage, and values close to 0 indicate that there is no effect of adding phage to the culture. Non-conditioned clones are those directly grown from glycerol stock, whereas pre-conditioned clones, had been reisolated twice, and grown overnight prior to performing the growth analyses. The ancestor, BJ1, sensitive, and the non-capsulated mutant (ΔwcaJ), are included as controls for the difference in culture conditions. Each small dot represents an independent biological replicate. Statistics represent t-tests to check for differences between clones directly from stock and those sequenced (after two passages in LB). ns means non-significative, *p<0.05, **p<0.01 and ***p<0.001.

The online version of this article includes the following figure supplement(s) for figure 6:

**Figure supplement 1.** Growth curves of resistant clones with no fixed mutations.

We then tested whether the lysogenized BJ1 clones, resistant to ST14 lysate, were also resistant to other lysates. To do so, we assessed plaque formation on lysogen lawns rather than growth inhibition, as these new lysogens already exhibited significant cell death due to phage outbursts (*Figure 3B*). Despite similarities between phages across lysates, BJ1 lysogens were only resistant to ST14 lysate, remaining sensitive to lysates produced by other strains (*Figure 7B*). Hence, and in contrast with the cross-resistance profile of clones with capsule modifications, integration of ST14 phages into BJ1 does not result in resistance to superinfection against a larger array of serotype-specific phages.

## Discussion

We explored how temperate phages drive population dynamics during coevolution between two *K. pneumoniae* strains under different degrees of parasite pressure. Based on theoretical works (*Brown et al., 2006*) and direct 24 hr competitions, we hypothesized that phage-sensitive BJ1 would coexist with polylysogenic ST14 in conditions with low phage infection but would be outcompeted fast in conditions of high phage concentration. We however observed that both ST14 and BJ1 are present in all populations. The varying relative frequency of the two strains over time suggests continuing co-evolution, since the initial drop in phage-susceptible BJ1 populations correlates with an increase in phage production, and then stabilization of both phage and BJ1 populations, prior to a second drop in BJ1 frequency. In populations evolving in the environment with MMC, BJ1 populations dropped faster than in the other conditions. This could be linked to a faster capsule inactivation.

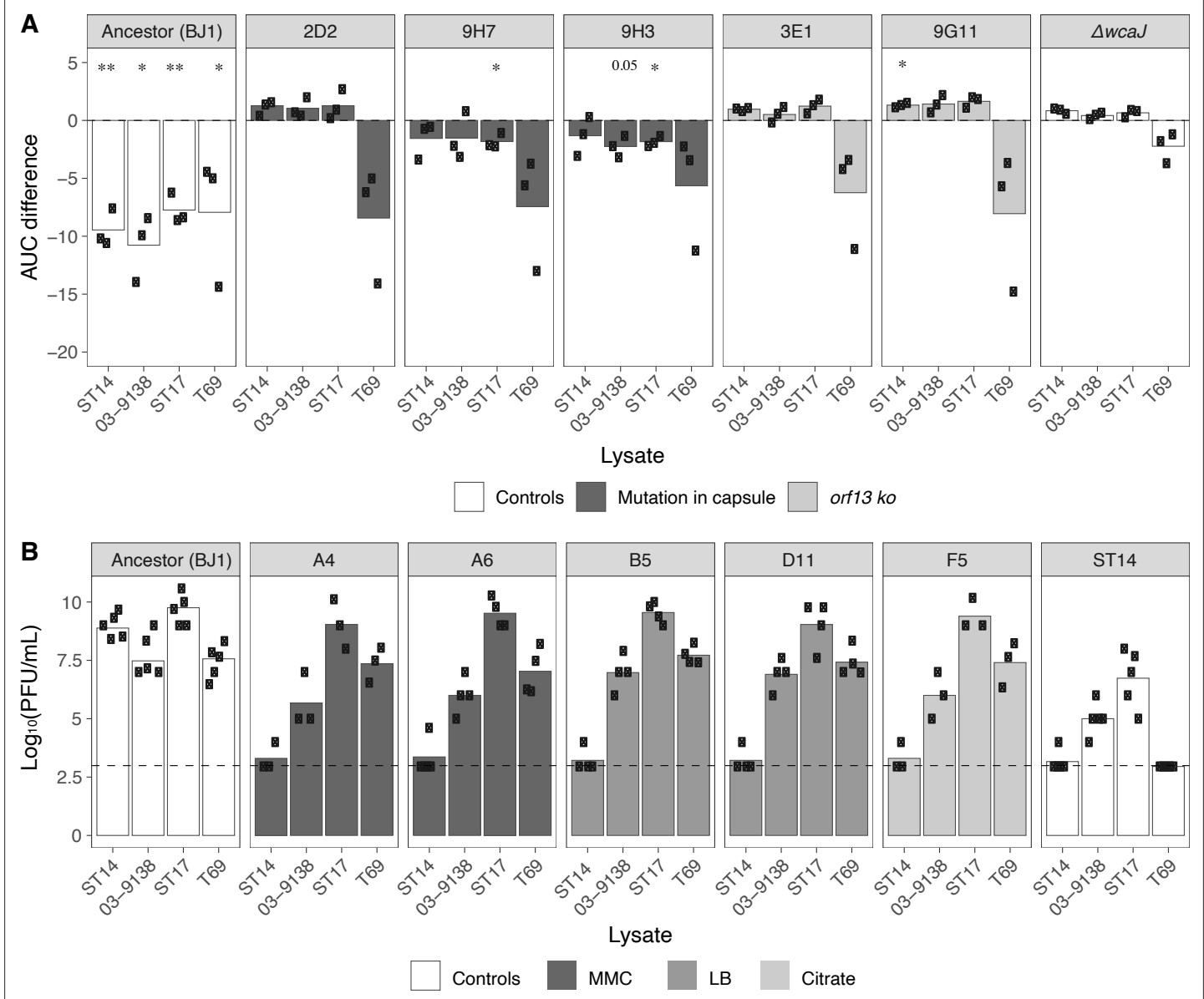

**Figure 7.** Cross-resistance to phages from other lysates. (**A**) The area under the curve (AUC) for capsulated (and non-lysogenized) resistant clones was calculated. The AUC of control cultures with LB was subtracted from those that were challenged with phage lysates. (Growth curves are shown in *Figure 7—figure supplement 1A*). Each dot represents an independent assay. One-sample t-tests were performed to test the difference from 0 (growth in LB). *p<0.05; **p<0.01 (**B**) PFU/mL of three independently generated lysates on lawns of new BJ1 lysogens from different evolutionary treatments. Lysogens A4 and A6 evolved in MMC and were isolated on day 1, lysogens B5 and D111 evolved in LB and were isolated on day 6 and 7, respectively, and clone F5 evolved in citrate and was isolated on day 7. Each black dot represents an independent biological replicate. The dashed line represents the limit of detection of the assay.

The online version of this article includes the following figure supplement(s) for figure 7:

**Figure supplement 1.** Characterization of lysates produced by other K2 strains.

Changes in the *Klebsiella* capsule are a primary mechanism of resistance to temperate phages, as shown for virulent phages (*Hesse et al., 2020*; *Tan et al., 2020*; *Cai et al., 2019*; *Majkowska-Skrobek et al., 2021*). More broadly, fast inactivation of the capsule confirms that alteration of the host recognition determinants frequently results in the protection of bacterial populations against phage (*Mutalik et al., 2020*; *Westra et al., 2015*; *Shaer Tamar and Kishony, 2022*). However, after six days of co-evolution, the frequencies of both strains in MMC reached an equilibrium. This also resulted in a higher frequency of BJ1 strain at the end of the evolution experiment, compared to

populations that were initially exposed to less phage pressure. High BJ1 frequencies in these populations are potentially linked to the emergence of diverse resistance mechanisms imposing lower fitness costs (*Figure 4*). Indeed, non-capsulated clones are fitter at the early stages of the coevolution, but capsulated non-lysogenized BJ1-resistant clones may outcompete other resistant BJ1 at longer evolutionary time scales.

Lysogenization often renders bacteria resistant to phages due to superinfection exclusion (*Cumby et al., 2012*; *Susskind et al., 1974*). Our previous work revealed frequent lysogenization of BJ1 when exposed to a highly concentrated lysate of ST14 (*de Sousa et al., 2020*), as expected from experimental (*Zeng et al., 2010*) and modeling studies (*de Sousa and Rocha, 2019*). We now show that lysogenization is not a stable mechanism of defense in this system (*Figure 3A*). This is because it results in high rates of spontaneous phage induction that lead to cell death. Similar observations were made with *E. coli* in a murine model in which high induction rates of phage lambda decreased the fitness of lysogens (*De Paepe et al., 2016*). *Frazão et al., 2022* also observed that after lysogenization, phage induction was very high, however, this was followed by progressive domestication, suggesting that the cost of some resistant mechanisms can be overcome. In the present work, the producing strain is a polylysogen, which could further complicate the acquisition of resistance since full protection requires multiple lysogenization events (one for each phage). For instance, lysogenic conversion was less frequent (~50% less) when a non-lysogenic *P. aeruginosa* was competed against a polylysogenic strain, compared to conditions when it was exposed to a single-lysogenized strain (*Burns et al., 2015*). Overall, lysogeny is disfavoured as a mechanism of phage resistance when the rates of spontaneous induction are high, when bacteria are targeted by multiple different phages, or when other less costly alternative resistant mechanisms may emerge.

A major finding of our work is that the dominant type of the resistance mechanisms changes across evolutionary time scales. At first, resistance emerged quickly to the loss of capsule production because functional inactivation can be achieved in many ways (*Chiarelli et al., 2020*; *Haudiquet et al., 2021*; *Nucci et al., 2022*; *Supplementary file 2*). At later stages, resistance may emerge by changes in the composition or expression of the capsule, which can outcompete both lysogens and non-capsulated clones. The emergence of these mechanisms later in the experiment suggests that such mutations are under a more limited supply than those resulting in capsule loss. This suggests that either long-term adaptation requires a small pool of rare mutations, contrary to gene inactivation that can be achieved in multiple ways, or that these adaptive paths are more complex, e.g., require multiple steps.

Surprisingly, our study revealed that some phage-resistant clones reverted to susceptibility after two passages in the absence of phage pressure. Yet, the sequence of these clones failed to reveal any mutations that could explain the phenotypic change. An increasing number of recent reports suggest that phage resistance could often be transient. For instance, Hesse and colleagues sequenced 57 different clones of *K. pneumoniae* resistant to a virulent phage and found that almost half of them lacked identifiable mutations (*Hesse et al., 2020*). Our clones could be transiently resistant because the nucleotide sequence of the gene *wcaJ* of the K2 capsule has short simple sequence repeats (SSR) whose changes are difficult to detect with the current genome assemblers and variant calling software. SSR are often mutational hotspots allowing rapid evolution of traits by changes that are easily reversible (*Moxon et al., 1994*). In agreement with this hypothesis, we found that capsule inactivation in ST14 is associated with an insertion of a thymine among a repeat of thymine residues (*Supplementary file 2*). Other, non-genetic mechanisms of transient resistance to phages have also been described (*Bull et al., 2014*; *Cota et al., 2015*; *Ongenae et al., 2022*; *Wohlfarth et al., 2023*). One is the epigenetic-based resistance based on DNA modifications, such as methylation. This was previously shown to regulate the length of the O-antigen length by phase variation in *Salmonella enterica*, and resulted in transient phage resistance (*Cota et al., 2015*). Additionally, transient cell wall shedding in filamentous actinobacteria, *B. subtilis* and *E. coli* (*Ongenae et al., 2022*) and conversion to cell wall-deficient L-forms in *Listeria monocytogenes* (*Wohlfarth et al., 2023*) can lead to transient phage resistance. An analogous process in *K. pneumoniae* could involve the shedding of the capsule in response to phage pressure, resulting in a state of reduced phage adsorption and limited infection. Such non-genetic protective changes are comparable to inducible immune responses (CRISPR-Cas) that incur in lower fitness costs relative to other constitutive changes such as permanent loss of receptors (*Westra et al., 2015*). Here, we show that at a longer evolutionary time scale, non-genetic modifications or inducible resistance can emerge. Contrary to a constitutive inactivation of the capsule,

non-genetic mechanisms do not result in added cost and the bacteria can revert to susceptibility once phage pressure is relieved.

Both the efficiency and fitness costs of phage-resistance mechanisms are context-dependent, as they can impose trade-offs in different environmental conditions (*Meaden et al., 2015*). We found that some mutants were only resistant to phages in liquid media, i.e., in the environment where they evolved, but remained fully sensitive to the same phages when growing on agar. This recapitulates phage-resistant *Pseudomonas syringae* evolved in vitro that did not display a fitness advantage compared to phage-sensitive cells when grown on a plant surface in the presence of phage (*Hernandez and Koskella, 2019*). These results are unexpected given previous studies showing that phage-sensitive bacteria are able to survive phage attacks in structured environments (agar, gut, or plant surface) due to the existence of refuges where sensitive cells may evade the phage (*Lourenço et al., 2020*; *Eriksen et al., 2018*; *Schrag and Mittler, 1996*; *Simmons et al., 2020*; *Testa et al., 2019*). However, survival in refuges is often contingent on the presence of other phage-resistant clones, or enhanced phenotypic resistance in structured environments due to reduced receptor expression (*Attrill et al., 2021*). A plausible explanation could be that in well-shaken liquid, phage-bacterium interactions are unstable, compared to interactions on agar (*Brockhurst et al., 2006*). If this is true, then less capsulated mutants could attain a higher level of resistance in liquid than in solid. Yet, resistance is not necessarily linked to a mere reduction in the production of the capsule because the Δ*rcsB* mutant that produces fewer capsule remains equally sensitive to phage in both liquid and agar (*Figure 5*). Taken together, our results highlight the complexity of the outcomes of bacterial interactions when they are affected by prophages.

## Materials and methods
### Bacterial strains and growth conditions
BJ1 (ENA: SAMEA4968482) and ST14 (ENA: SAMN22024794) are two phylogenetically-distant *K. pneumoniae*, both bearing the K2 capsule serotype. Bacteria were grown at 37° in Luria-Bertani (LB) agar plates or in 4 mL of liquid broth under vigorous shaking (250 rpm). Chloramphenicol (30 µg/ml) and trimethoprim (100 µg/ml) were used to select for strain BJ1 and ST14, respectively.

### Competition calculations
Calculations of $B_{ij}$, $C_i(j)$ were performed as reported in *Fiegna and Velicer, 2005*. (*i*) *Unidirectional mixing-effect parameter $C_i(j)$*. The effect of mixing two strains *i* and *j* on the population yield during the growth of focal strain *i* was quantified by the one-way mixing effect parameter $C_i(j)$. To calculate this parameter, the expected log10-transformed yield of strain *i* based on pure-culture performance (corrected for the frequency at which strain *i* was added) was subtracted from its actual log-transformed yield during competition with strain *j*.

$$C_i\left(j\right) = \log10\left(\frac{N_i\left(j,\ t24\right)}{N_i\left(j,\ t0\right)}\right)\log10\left(\frac{N_i\left(t24\right)}{N_i\left(t0\right)}\right)$$

Positive $C_i(j)$ values indicate that strain *i* grew to a higher population size in the experiments in the presence of strain *j* than in pure culture, whereas a negative value indicates that mixing with *j* negatively affected the growth yield of *i*. (*ii*) *Bidirectional mixing effect parameter $B_{ij}$*. $B_{ij}$ is the difference between the actual total (log₁₀-transformed) group cell count in a mix of strains *i* and *j* and the value expected from pure culture performance of the same strains (corrected for initial frequencies of each strain).

$$B_{ij} = \log\left(\frac{N_i\left(j,\ t24\right) + N_j\left(i,\ t24\right)}{\frac{N_i\left(t24\right)}{2} + \frac{N_j\left(t24\right)}{2}}\right)$$

Positive and negative $B_{ij}$ values indicate that total productivity is higher or lower, respectively, than expected from pure culture performance.

## Coevolution experiment

Three clones from each strain were used to inoculate overnight cultures, which were then diluted at 1:100 and used to initiate the three independent mixed populations in a ratio 1:1, in a final volume of 4 mL. Each of the three mixed populations evolved in three different environments: (i) LB, (ii) LB supplemented with 0.2% citrate, and (iii) LB with mitomycin C (MMC, 0.1 µg/mL). Cultures were allowed to grow for 24 hr at 37 °C and diluted again to 1:100 in fresh media. This was repeated for 30 days. Each day, each independently evolving population was plated and serially diluted. CFUs were counted (three plates per sample) and the emergence of non-capsulated mutants was recorded. Non-capsulated mutants are easily visualized by the naked eye as mutants produce smaller, rough, and translucent colonies.

## Phage experiments

(*i*) *Growth curves:* 200 µL of diluted overnight cultures of *Klebsiella spp.* (1:100 in fresh LB) were distributed in a 96-well plate. Cultures were allowed to reach OD = 0.2 and either mitomycin C to 1 µg/mL or 20 µl of PEG-precipitated induced and filtered supernatants at $2 \times 10^8$ PFU/mL were added. Growth was then monitored until the late stationary phase. (*ii*) *PEG-precipitation of phages*. Overnight cultures were diluted 1:500 in fresh LB and allowed to grow until OD = 0.2. Mitomycin C was added to the final 5 µg/mL. After 4 hr at 37 °C, cultures were centrifuged at 4000 rpm and the supernatant was filtered through 0.22 µm. Filtered supernatants were mixed with chilled PEG-NaCl 5 X (PEG 8000 20% and 2.5 M of NaCl) and mixed through inversion. Phages were allowed to precipitate for 15 min and pelleted by centrifugation for 10 min at 13,000 rpm at 4 °C. The pellets were dissolved in TBS (Tris Buffer Saline, 50 mM Tris-HCl, pH 7.5, 150 mM NaCl). (*iii*) *Calculating plaque forming units (PFU)*. Overnight cultures of susceptible or tested strains were diluted 1:100 and allowed to grow until OD = 0.8. 250 µL of bacterial cultures were mixed with 3 mL of top agar (0.7% agar) and poured into prewarmed LB plates. Plates were allowed to dry before spotting serial dilutions of induced PEG-precipitated phages. Plates were left overnight at room temperature and phage plaques were counted. (*iv*) *Phage adsorption*. Adsorption of phage particles to the cell surface was performed as previously described (**Hesse et al., 2020**). Briefly, each resistant clone was grown until OD ~0.35. One ml of each culture was transferred to separate wells in a 24-well plate, to which 10 µl of filtered phage lysate (*ca*. $5*10^6$ phage particles) was added. The mix was allowed to sit for 2 min at room temperature prior to incubation at 37 °C for 15 min with shaking at 140 rpm. Phage adsorption was measured by quantifying the free phage remaining in the solution, after centrifugation for 10 min at 10,000 rpm, to get rid of bacterial cells. The supernatant was serially diluted and the non-adsorbed phage was quantified by spot titer on a bacterial lawn of strain BJ1. Finally, to quantify how many phage was adsorbed, the non-adsorbed phage was subtracted from the initial amount of phage added to the culture.

## Sequencing

(*i*) *Genomes of phage-resistant clones*. Single clones were allowed to grow overday in LB supplemented with 0.7 mM EDTA, to limit capsule production. We performed DNA extraction with the guanidium thiocyanate method, with few modifications (**Pitcher et al., 1989**). RNAse A treatment (37 °C, 30 min) was performed before DNA precipitation. Each clone (n=15) was sequenced by Illumina with 150pb paired-end reads, yielding approximately 1 Gb of data per clone. The reads were compared to the reference genome using *breseq* v0.33.2, default parameters. (*ii*) *wcaJ gene*. PCR of wcaJ was performed using the primers that hybridized 150 base pairs upstream and downstream of the wcaJ gene; K2.wcaJ.150–5 (5′- GGCGTTCCAGCAAGGGTTATC –3′) and K2.wcaJ.150–3 (5′-ACGT TCGCGCTTAAATGTG-3′), respectively. To allow full coverage of the gene, PCR products were also sequenced with primer K2.wcaJ.inseq-5 (5′-CTGGGTCTTTACAGAGGAATC-3′). PCR products were sequenced by Sanger and analyzed using *APe*.

## Capsule quantification

The bacterial capsule was extracted as described in **Domenico et al., 1989**. Briefly, 500 µL of an overnight culture was adjusted to OD of two and mixed with 100 µL of 1% Zwittergent 3–14 detergent in 100 mM citric acid (pH 2.0) and heated at 56 °C for 20 min. Afterward, it was centrifuged for 5 min at 14,000 rpm and 300 µL of the supernatant was transferred to a new tube. Absolute ethanol was

added to a final concentration of 80% and the tubes was placed on ice for 20 min. After a second wash with ethanol at 70%, the pellet was dried and dissolved in 250 µL of distilled water. The pellet was then incubated for 2 hr at 56 °C. Polysaccharides were then quantified by measuring the amount of uronic acid, as described in *Blumenkrantz and Asboe-Hansen, 1973*. A 1,200 µL volume of 0.0125 M tetraborate in concentrated $H_2SO_4$ was added to 200 µL of the sample to be tested. The mixture was vigorously vortexed and heated in a boiling-water bath for 5 min. The mixture was allowed to cool, and 20 µL of 0.15% 3-hydroxydiphenol in 0.5% NaOH was added. The tubes were shaken, and 100 µL were transferred to a microtiter plate for absorbance measurements (520 nm). The uronic acid concentration in each sample was determined from a standard curve of glucuronic acid.

## Citrate quantification

24 hr cultures of BJ1, ST14, their coculture, or blank tubes with citrate, were centrifuged for 10 min at 4000 rpm and the supernatant was sterilized with 0.22 µm filter, prior to deproteination by centrifugation in an Amicon tube (10 kDa). Citrate concentration was measured using a Citrate assay kit (Sigma-Aldrich MAK333).

## Individual-based simulations of bacteria-phage interactions

Simulations were performed based on the model described in *de Sousa and Rocha, 2019*. Briefly, both bacterial cells and phage particles are independent individuals in an environment represented as a two-dimensional grid. The environment is simulated as well-mixed, meaning that positions of bacteria and phage are randomized at each iteration. Bacterial death can be intrinsic (e.g. of old age) or explicit (e.g. lysed by phage). Bacteria can resist phage infection by acquiring a mutation that mimics capsule loss (at varying rates, with a varying fitness cost, see results). Upon phage infection, the phage can either follow a lytic cycle or a lysogenic one, according to a stochastic decision defined by the parameters LysogenyAlpha and LysogenyKappa, which takes into consideration the density of nearby phages. When lysogenized, bacteria become insensitive to new phage infections, but the integrated prophage can excise (and thus lead to the death of this specific cell) at varying frequencies (see results). The simulations we explored are initiated with 10,000 bacterial cells, and 1000 phage particles are added into the environment at the beginning of the simulation. For each condition explored (varying the probability of phage induction, the probability of bacteria acquiring a phage resistance mutation, and the cost of this mutation), we performed 30 replicate simulations, each running for 150 iterations. The values presented in the results correspond to the median of the 30 replicate simulations, for each condition. The set of parameters explored, that are relevant to the questions in this study, are shown in Text S1. Other mechanisms that can be simulated in eVIVALDI (e.g. transduction) were not used in these simulations.

## wGRR calculations

Phage similarity was calculated as described in *de Sousa et al., 2020*. Briefly, we searched for sequence similarity between all proteins of all phages using mmseqs2 (*Steinegger and Söding, 2017*) with the sensitivity parameter set at 7.5. The results were filtered with the following parameters: e-value lower than 0.0001, at least 35% identity between amino acids, and a coverage of at least 50% of the proteins. The filtered hits were used to compute the set of bi-directional best hits (bbh) between each phage pair. This was then used to compute a score of gene repertoire relatedness for each pair of phage genomes, weighted by sequence identity, computed as follows:

$$\text{wGRR}_{A,B} = \sum_i \frac{id(A_i, B_i)}{\min(A, B)}$$

where $A_i$ and $B_i$ is the pair $i$ of homologous proteins present in $A$ and $B$ (containing respectively #A and #B proteins), $id(A_i, B_i)$ is the percent sequence identity of their alignment, and $\min(A, B)$ is the total number of proteins of the smallest prophage, i.e., the one encoding the smallest number of proteins ($A$ or $B$). wGRR varies between zero and one. It amounts to zero if there are no orthologs between the elements, and one if all genes of the smaller phage have an ortholog 100% identical in the other phage. Hence, the wGRR accounts for both frequencies of homology and the degree of similarity among homologs.

## Acknowledgements

The sequencing work was made at the Biomics Platform, C2RT, Institut Pasteur, Paris, France, supported by France Génomique (ANR-10-INBS-09) and IBISA. We thank Alex Hall for critical reading of the manuscript. We would also like to thank the reviewers for their contribution which critically improved the manuscript. This work was funded by an ANR JCJC (Agence national de recherche) grant [ANR 18 CE12 0001 01 ENCAPSULATION] awarded to OR. The laboratory is funded by a Laboratoire d'Excellence 'Integrative Biology of Emerging Infectious Diseases' (grant ANR-10-LABX-62-IBEID) and the FRM [EQU201903007835]. The funders had no role in study design, data collection and interpretation, or the decision to submit the work for publication.

## Additional information

### Funding

| Funder | Grant reference number | Author |
| --- | --- | --- |
| Agence Nationale de la Recherche | ANR 18 CE12 0001 01 ENCAPSULATION | Olaya Rendueles |
| Agence Nationale de la Recherche | ANR-10-LABX-62-IBEID | Eduardo PC Rocha |
| Fondation pour la Recherche Médicale | EQU201903007835 | Eduardo PC Rocha |

The funders had no role in study design, data collection and interpretation, or the decision to submit the work for publication.

### Author contributions

Olaya Rendueles, Conceptualization, Resources, Data curation, Formal analysis, Supervision, Funding acquisition, Investigation, Visualization, Methodology, Writing - original draft, Project administration, Writing – review and editing; Jorge AM de Sousa, Software, Formal analysis, Investigation, Writing – review and editing; Eduardo PC Rocha, Resources, Supervision, Funding acquisition, Project administration, Writing – review and editing

### Author ORCIDs

Olaya Rendueles http://orcid.org/0000-0002-6648-1594
Eduardo PC Rocha http://orcid.org/0000-0001-7704-822X

### Decision letter and Author response

Decision letter https://doi.org/10.7554/eLife.83479.sa1
Author response https://doi.org/10.7554/eLife.83479.sa2

## Additional files

### Supplementary files

• Supplementary file 1. PHASTER prophage prediction in strain ST14. The genome was analyzed with PHASTER (*Arndt et al., 2016*) in April 2021.

• Supplementary file 2. Mutations accumulated in *wcaJ* gene resulting in non-capsulated mutants. ND; none detected. On the day in which at least 50% of the clones of each independently evolving population were non-capsulated, two of such clones were isolated (# Clone), the *wcaJ* gene amplified by PCR and sequenced by Sanger. For BJ1, this was repeated twice independently (# Seq) (N=36 for BJ1 and N=18 for ST14). Independently evolving populations (Pop) are identified with a number from 1 to 3. Sequencing of the ancestor revealed that no mutations were present in the *wcaJ* genes.

• MDAR checklist

## Data availability

All raw data is available on figshare.com; https://doi.org/10.6084/m9.figshare.22101998.

The following dataset was generated:

| Author(s) | Year | Dataset title | Dataset URL | Database and Identifier |
|---|---|---|---|---|
| Rendueles O, deSousa JAM, Rocha EPC | 2023 | Competition between lysogenic and sensitive bacteria is determined by the fitness costs of the different emerging phage-resistance strategies | https://doi.org/10.6084/m9.figshare.22101998.v2 | figshare, 10.6084/m9.figshare.22101998.v2 |

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
