## [Editor Report]

The overarching question of the manuscript is important and the findings inform the patterns and mechanisms of phage-mediated bacterial competition, with implications for microbial evolution and antimicrobial resistance. The strength of the evidence in the manuscript is compelling, with a huge amount of data and very interesting observations. The conclusions are well supported by the data. This manuscript provides a new co-evolutionary perspective on competition between lysogenic and phage-susceptible bacteria, that will inform new studies and sharpen our understanding of phage-mediated bacterial co-evolution.

---

## [Decision Letter]

**Decision letter after peer review:**

Thank you for submitting your article "Competition between lysogenic and sensitive bacteria is determined by the fitness costs of the different emerging phage-resistance strategies" for consideration by *eLife*. Your article has been reviewed by 2 peer reviewers, including Samuel L Díaz-Muñoz as the Reviewing Editors and Reviewer #2, and the evaluation has been overseen by Christian Landry as the Senior Editor.

Essential revisions:

1) The manuscript concludes that selection favors more subtle capsule mutations because they are less costly than capsule-loss mutants (lines 497-500). However, there are no data to support this conclusion, as fitness costs of the various resistance phenotypes analyzed were not measured.

The revised manuscript should either include such data if available or, without such data, substantial changes in the focus of the paper would have to be made, as the manuscript places considerable emphasis on the importance of fitness costs in resistance evolution, notably in the title.

2) The manuscript attempts to directly relate the emergence of the various resistance mechanisms to phage infection pressure during the coevolution experiment. However, while bacterial population dynamics data are monitored meticulously, the reviewers were not able to find the phage dynamics data. It seems likely to the reviewers that these data exist and these data are needed to make conclusions that link resistance mechanisms to phage pressure.

In the absence of these data, a major change in the narrative would be needed to bring the conclusions of the paper in line with the evidence (i.e. conclusions would be related to competition with the lysogenic strain and avoid a mention of phage pressure).

3) We note that if there is no further evidence to support both conclusions noted in 1) and 2) above, then the conclusions made by the manuscript would be substantially altered and the significance of the contribution would also substantially change.

*Reviewer #1 (Recommendations for the authors):*

The authors have gathered a huge amount of data and made some very interesting observations but unfortunately, a clear narrative is missing and the data do not support the main conclusions. One relatively simple thing would be to measure phage titers alongside bacterial densities over time – this would allow the authors to directly say something about phage infection pressure. To understand whether resistance mechanisms confer different fitness costs in the presence/absence of phage (and high/low phage pressure) should in my view be tested separately and directly in short-term experiments by directly competing clones with different resistance mechanisms isolated from the experiment. If the latter is not possible (genetic markers may not be available to tease them apart easily – qPCR is a good alternative but obviously more labour-intensive and expensive) the authors could measure bacterial growth curves instead to make inferences about their fitness.

*Reviewer #2 (Recommendations for the authors):*

I think this is a very well-done and very interesting study! Please see below my recommendations:

1) Rearrange the discussion to provide the salient overall, general points. Make it. I think those major take-home messages are that 1) lysogenization might not be as prevalent a phage resistance mechanism as we thought previously (or at least in as wide a set of conditions as we thought), particularly in the long-term and 2) that receptors continue to be the primary mechanism, importantly through loss at first and then through changes in expression/composition. Secondarily, the multiple mechanisms and the transience can be discussed, but as written these aspects detract from the overall message.

2) Please acknowledge previous work on context dependence (i.e. agar/liquid) from multiple groups, but Britt Koskella and Ellie Harrison come to mind immediately.

---

## [Author Response]

Essential revisions:1) The manuscript concludes that selection favors more subtle capsule mutations because they are less costly than capsule-loss mutants (lines 497-500). However, there are no data to support this conclusion, as fitness costs of the various resistance phenotypes analyzed were not measured.The revised manuscript should either include such data if available or, without such data, substantial changes in the focus of the paper would have to be made, as the manuscript places considerable emphasis on the importance of fitness costs in resistance evolution, notably in the title.

We now provide a novel figure in the main text, Figure 4, comparing the growth of all isolated clones with different resistance mechanisms, in the presence and absence of phages, and at different periods during the evolution experiment. Our data further support our conclusion that new lysogens are less fit than other resistant clones.

2) The manuscript attempts to directly relate the emergence of the various resistance mechanisms to phage infection pressure during the coevolution experiment. However, while bacterial population dynamics data are monitored meticulously, the reviewers were not able to find the phage dynamics data. It seems likely to the reviewers that these data exist and these data are needed to make conclusions that link resistance mechanisms to phage pressure.In the absence of these data, a major change in the narrative would be needed to bring the conclusions of the paper in line with the evidence (i.e. conclusions would be related to competition with the lysogenic strain and avoid a mention of phage pressure).

We agree with the reviewer that including phage titers is important.

In light of these comments, we provide a new supplementary figure (Figure S4) showing that phages are being produced in large numbers in our coevolution setting until the end of the experiment. This strongly suggests that evolutionary outcomes are still affected by the presence of phage and not solely caused by clonal competition.

3) We note that if there is no further evidence to support both conclusions noted in 1) and 2) above, then the conclusions made by the manuscript would be substantially altered and the significance of the contribution would also substantially change.

We hope that the modifications we have made to address points (1) and (2) prove that our conclusions are fully supported by our data.

Reviewer #1 (Recommendations for the authors):The authors have gathered a huge amount of data and made some very interesting observations but unfortunately, a clear narrative is missing and the data do not support the main conclusions. One relatively simple thing would be to measure phage titers alongside bacterial densities over time – this would allow the authors to directly say something about phage infection pressure.

See answer to Editor’s comment #2.

The new supplementary figure S4 shows phage titers at three different time points of the experiment (days 10, 20 and 30) indicating that phages are being produced in large numbers in our coevolution setting until the end of the experiment. This strongly suggests that evolutionary outcomes are, at least partially, influenced by the presence of phage and not solely due to clonal competition.

To understand whether resistance mechanisms confer different fitness costs in the presence/absence of phage (and high/low phage pressure) should in my view be tested separately and directly in short-term experiments by directly competing clones with different resistance mechanisms isolated from the experiment. If the latter is not possible (genetic markers may not be available to tease them apart easily – qPCR is a good alternative but obviously more labour-intensive and expensive) the authors could measure bacterial growth curves instead to make inferences about their fitness.

See answer to Editor’s comment #1.

We now provide an extended analysis deriving from the growth curves from all clones isolated during the experiment, in the presence and absence of phage. The use of genetic markers or performing qPCRs would have reduced the analysis to only a few resistant clones -*i.e.* those for which we already know the genetic basis of the resistance-. As expected, our results show that different mechanisms of resistance have different fitness costs, but these may change with time. This is now addressed in a new section of the results.

Reviewer #2 (Recommendations for the authors):I think this is a very well-done and very interesting study! Please see below my recommendations:1) Rearrange the discussion to provide the salient overall, general points. Make it. I think those major take-home messages are that 1) lysogenization might not be as prevalent a phage resistance mechanism as we thought previously (or at least in as wide a set of conditions as we thought), particularly in the long-term and 2) that receptors continue to be the primary mechanism, importantly through loss at first and then through changes in expression/composition. Secondarily, the multiple mechanisms and the transience can be discussed, but as written these aspects detract from the overall message.

We have rewritten our discussion to better emphasize our key findings and streamline our thoughts, in light of the reviewer’s comment. Specifically, we better highlight that lysogenization is not as prevalent as previously thought in longer time scales. As suggested, we also discuss the loss of receptor as a defence mechanisms, but we chose to also focus on the emergence of transient changes (phenotypic resistance) as we believe (and provide large evidence from recent publications) that it is a wide-spread resistance mechanism, and very often neglected.

2) Please acknowledge previous work on context dependence (i.e. agar/liquid) from multiple groups, but Britt Koskella and Ellie Harrison come to mind immediately.

We have expanded our discussion to cite other previous works more including references by the mentioned scientists.